# Plasmonic Nanostructures for Broadband Solar Absorption Based on Synergistic Effect of Multiple Absorption Mechanisms

**DOI:** 10.3390/nano12244456

**Published:** 2022-12-15

**Authors:** Junli Su, Dingquan Liu, Leihao Sun, Gang Chen, Chong Ma, Qiuyu Zhang, Xingyu Li

**Affiliations:** 1Shanghai Institute of Technical Physics, Chinese Academy of Sciences, Shanghai 200083, China; 2School of Physical Science and Technology, ShanghaiTech University, Shanghai 200031, China; 3School of Optoelectronics, University of Chinese Academy of Sciences, Beijing 100049, China

**Keywords:** solar energy, broadband solar absorption, magnetic polariton, light trapping, optical resonator

## Abstract

The growing attention to solar energy has motivated the development of highly efficient solar absorbers. In this study, a high-performance meta-structure solar absorber (MSSA) based on a tungsten truncated cone structure combined with a film resonator structure has been proposed and demonstrated numerically. The designed structure exhibits over 97.1% total solar absorption efficiency and less than 8.5% total thermal emissivity under the condition of one solar concentration, hence reaching 91.6% photothermal conversion efficiency at 100 °C. In addition, the proposed MSSA achieves promisingly high spectrum absorptance of over 97.8% in the ultraviolet, visible and near-infrared regions (280–1700 nm). Based on the simulation analysis, the enhanced light absorption is attributed to the synergistic effect of the magnetic polaritons (MPs) on the nanostructured metal surface, the cavity plasmon resonance between the truncated cones that can form light-trapping structures, the magnetic field resonance of the metal–insulator–metal (MIM) optical resonator and the inherent loss of tungsten. The impedance of the absorber is well matched with free space. Furthermore, the optimized absorber shows great potential in solar thermophotovoltaic applications that require wide-angle polarization-independent ultra-broadband light response characteristics.

## 1. Introduction

The sharply rising price of fossil energy and its harmful impact on the environment force society to use clean energy, and solar energy is regarded as the most promising and abundant renewable source of clean energy on Earth. Therefore, it is crucial to investigate broadband solar absorbers, which can be widely used in photocatalysis [1,2], solar cells [3,4], photodetection [5,6], thermionics (TE) [7], photon-enhanced thermionic emission (PETE) [8], solar steam generators [9,10] and solar thermophotovoltaics [11,12], to improve the efficiency of converting solar energy. Various approaches have been proposed and utilized, including the destructive interference of metal dielectric stacks [13,14], intrinsic absorption of semiconductors [15,16], plasmonic near-field strength absorption of cermet [17,18] and light-trapping enhancement absorption of surface-textured “black” film [7,19]. Some absorbers have been commercialized and applied; however, challenges such as wide-angle absorption and poor solar energy conversion efficiency at high temperature still remain. Recently, advances in nanoscale fabrication techniques have led to the rise of metamaterial solar absorbers, which exhibit perfect electromagnetic responses in specific regions of wavelength. However, the perfect electromagnetic response only emerges in a narrow band range because of the size effect of nanopatterns in the metamaterial absorbers [20]. Meanwhile, due to atmospheric scattering and the Earth’s rotation, it is necessary to consider the non-polarized and wide-angle nature of incident sunlight. The ideal broadband solar absorbers need to achieve the absorption of an arbitrary polarization state and wide-angle incident solar radiation to improve the conversion efficiency of solar energy.

Metals are the most common surface plasmon materials, and noble metal materials such as gold and silver are widely used in the nanoparticles or nano-resonators of periodic structure absorbers, which have perfect optical responses in the narrow band of visible light [21,22]. However, gold and silver are too expensive, and the imaginary part of the dielectric constant is low. In addition, the broadband absorber needs to improve its response in the near-infrared region, and the electron–electron scattering during noble metal plasmon resonance also produces excessive Joule heat, which results in an enhancement in thermal emissions and a reduction in energy conversion efficiency. Conversely, this could be an advantage for absorbers coupled to thermionic emitters (TE and PETE), because enhanced thermal emissions may result in higher heat-to-current conversion efficiency [8]. By contrast, the stability of plasmonic absorbers of ordinary metals is weak. Thus, it is imperative to find low-cost materials with strong thermal stability. Tungsten is a refractory metal with extremely high thermal stability, since the melting point of tungsten is around 3400 °C. Moreover, the strong plasmon resonance of tungsten is caused by the high imaginary part of the permittivity due to the interband transition, and the optical response is significantly redshifted compared with gold and silver [23,24]. Hence, tungsten can serve as a plasmon metal during high-power operation.

In this work, we proposed an MSSA composed of tungsten truncated cones and a tungsten–alumina–tungsten (W-Al_2_O_3_-W) MIM structure, which utilized the synergistic effect of the excitation of MPs on the tungsten metal surface, the light-trapping structures formed by cavity plasmon resonance between truncated cones, the magnetic field resonance of the MIM optical resonator and the inherent loss of metal tungsten (W) to achieve high spectrum absorptance of over 97.8% in the ultraviolet, visible and near-infrared regions (280–1700 nm). Furthermore, the photothermal conversion efficiency was calculated to quantitatively evaluate the performance of the designed solar absorber. The essential mechanisms responsible for the absorptance enhancement were revealed and demonstrated in detail. Additionally, the geometrical effects on the spectral absorption properties of the absorber were further analyzed. Finally, omnidirectional and polarization-independent optical absorption properties were also demonstrated in this absorber. Our work paves the way to the design of multiple-exciting energy-harvesting devices, which will have potential in the field of high-temperature-resistant concentrated solar power applications (CSP).

## 2. Structure Design and Simulation Method

Figure 1 shows a 3D schematic diagram and a cross-sectional view of the unit cell structure of the proposed MSSA array, which consists of truncated cones on the top and an MIM thin film structure beneath. The dielectric material is alumina, and the metal material is tungsten, which has a high melting point, good corrosion resistance and abundant reserves. The proposed solar absorber can be fabricated with vacuum coating equipment and electron beam lithography (EBL) or a femtosecond laser [25,26]. Specifically, tungsten and alumina layers can be, respectively, deposited by magnetron sputtering and reactive magnetron sputtering; then, the surface structures can be fabricated by electron beam etching, or using a tightly focused femtosecond laser beam. The laser-induced periodic surface structure (LIPSS) treatment may not require necessarily a vacuum and does not need time-consuming technological steps, as with EBL [27,28]. As shown in Figure 1b, the structural geometric parameters include the diameter of the upper surface circle of the truncated cone $d_{1}$, the diameter of the lower surface circle $d_{2}$, the height of the tungsten truncated cone h, the thickness of the upper tungsten layer $t_{1}$ in the MIM film structure, the thickness of the Al_2_O_3_ dielectric layer $t_{2}$, the thickness of the lower tungsten layer (i.e., substrate) $t_{3}$ and the period P. The MSSA in Figure 1 was designed and calculated via FDTD Solutions of the LUMERICAL Corporation [29]. In this study, the optical constants of tungsten and alumina are taken from Palik’s book [30]. Moreover, the temperature dependence of optical constants is ignored in the simulated calculation and analysis.

The optical properties cannot be simply expressed by the optical parameters of the constituent materials in the electromagnetic metamaterial, so the effective medium theory is often used to calculate the overall effective parameters of the equivalent structure. In this paper, S-parameters are extracted to represent the overall optical response and effective impedance of the superstructure when using the three-dimensional finite time domain difference method to solve the partial differential equation system based on Maxwell’s equations [31,32]. The absorptance of the whole structure can be calculated by
(1)A(ω)=1−R(ω)−T(ω)
where A(ω), R(ω) and T(ω) are the spectral absorbtance, reflectance and transmittance, respectively. For metamaterial absorbers with high symmetry in the direction perpendicular to the propagation direction of light, the reflectance and transmittance can be expressed briefly:(2)$$R\left(\omega \right)=\left|S_{11}\right|=\left|S_{22}\right|$$
(3)$$T\left(\omega \right)=\left|S_{21}\right|=\left|S_{12}\right|$$

$S_{11}$ and $S_{22}$, $S_{21}$ and $S_{12}$ are the reflection coefficient and transmission coefficient of the scattering matrix under normal incidence conditions, respectively. Additionally, the thickness of the substrate is set to 200 nm, which far exceeds the skin depth of visible and near-infrared light in metal materials. Therefore, T(ω)=0, and Equation (1) can be simplified to
(4)$$A\left(\omega \right)=1-\left|S_{11}\right|$$

The effective impedance $Z_{eff}\left(\omega \right)$ of the metamaterial can be expressed by the following formula [33]:(5)$$Z_{eff}\left(\omega \right)=\sqrt{\frac{{\left(1+S_{11}\right)}^{2}-S_{21}{}^{2}}{{\left(1-S_{11}\right)}^{2}-S_{21}{}^{2}}}$$

Under solar irradiation, the perfect selective absorber should have perfect absorption in the ultraviolet, visible and near-infrared bands (i.e., 0.28–2.5 μm) and zero emission in the mid-infrared thermal radiation band (i.e., 2.5–30 μm). The performance of the metamaterial absorber can be represented by the total solar radiation absorptivity $\alpha_{total}$ and the total thermal emissivity $\varepsilon_{total}$ at normal incidence, which are expressed as follows [13,34,35]:(6)$$\alpha_{total}=\frac{\int_{0.28\mu m}^{2.5\mu m}A\left(\lambda \right)I_{S}\left(\lambda \right)d\lambda }{\int_{0.28\mu m}^{2.5\mu m}I_{S}\left(\lambda \right)d\lambda }$$
(7)$$\varepsilon_{total}=\frac{\int_{2.5\mu m}^{30\mu m}\;\left(\lambda \right)I_{B}\left(\lambda ,T\right)d\lambda }{\int_{2.5\mu m}^{30\mu m}I_{B}\left(\lambda ,T\right)d\lambda }$$
where A(λ) is the spectral absorptance of the solar absorber, which can be obtained from the reflectance spectrum of the absorber, i.e., A(λ)=1−R(λ). $I_{S}\left(\lambda \right)$ represents the solar radiation power density taken from the global tilt AM 1.5 data, which was measured in the United States [36]. In detail, solar radiation will be absorbed and scattered by atmospheric components and thus attenuated when the light passes through the atmosphere. Light with a wavelength less than 300 nm will be absorbed by oxygen (O_2_), ozone (O_3_) and nitrogen (N_2_), and infrared light will be mainly absorbed by water vapor (H_2_O) and carbon dioxide (CO_2_). This decay can be described by the air mass (AM), which is a dimensionless number, defined as AM=1/cosθ, where θ is the angle between the incident light and the ground normal [37]. AM1.5 denotes the corresponding angle between the incident light and the ground normal, which is 48.2°. According to Planck’s law, $I_{B}\left(\lambda ,T\right)$ represents the black body radiation power density of the absorber at temperature T, and ε(λ) is the spectral emittance of the absorber at this temperature. In our design, the absorber is opaque, not only due to the 200-nm-thickness substrate, but also because the perfectly matched layer (PML) boundary condition was set during the simulation. According to Kirchhoff’s law of thermal radiation, the spectral emittance of the absorber is equal to the absorptance, i.e., ε(λ)=A(λ), when the MSSA is in thermal equilibrium with the environment [38].

According to the Stefan–Boltzmann law, when ignoring the heat loss caused by conduction and convection, the overall photothermal conversion efficiency of the absorber under solar radiation intensity is
(8)$$\eta =\alpha_{total}-\varepsilon_{total}\frac{\sigma \left(T_{abs}^{4}-T_{air}^{4}\right)}{CI_{S}}$$
where $\sigma =5.67\times 10^{-8}\;W\;\cdot \;m^{-2}\;\cdot \;K^{-4}$ is the Stefan–Boltzmann constant, C is the solar concentration factor (i.e., the ratio of gathered intensity via solar-tracking mirrors to the solar intensity), $I_{S}$ is the solar irradiance at AM1.5 G, which is around $1000\;W\;\cdot \;m^{-2}$ [38], and $T_{abs}$ and $T_{air}$ are the temperatures of the absorber and surroundings.

## 3. Results and Discussion

### 3.1. Meta-Structure Absorber for Solar Energy Harvesting

The optimal structural parameters of the MSSA are as follows: $d_{1}=70\;\mathrm{nm}$, $d_{2}=400\;\mathrm{nm}$, h=760 nm, $t_{1}=10\;\mathrm{nm}$, $t_{2}=90\;\mathrm{nm}$ and P=425 nm. $t_{3}$ is the thickness of the substrate and was set to 200 nm. Figure 2a shows the reflectance spectra of the MSSA under normal incidence of transverse electric (TE) polarized light (electrical field in x-direction), the solar radiation flux density at different solar concentration factors and the infrared radiation density of the absorber at different temperatures, respectively. The exploitation of the condenser lens will increase the unit energy density received by the absorber, thereby increasing the temperature of the absorber. Ultimately, the rise in temperature makes the peak energy density of the blackbody radiation blueshift with the wavelength. In order to balance the total solar radiation absorptivity $\alpha_{total}$ and the total thermal emissivity $\varepsilon_{total}$, as well as maintain the best photothermal conversion efficiency η, the cut-off wavelength of the absorber [$R\left(\lambda_{cut-off\;wavelength}\right)=0.5$] will also be blueshifted. Thus, when designing the absorber, different structural parameters can be adjusted to control the cut-off wavelength to correspond to different application conditions. As depicted in Figure 2a, the purple line shows the near-perfect absorption of MSSA in the ultraviolet, visible and near-infrared regions. For instance, the spectral reflectance of the absorber is less than 2.2% in the wavelength region of 280–1700 nm, and less than 10% in the wavelength region of 280–1800 nm. In the infrared region above 3000 nm, the spectral reflectance remains above 90%, reflecting the low thermal emissivity. In addition, as shown in Figure 1, the MSSA has a high degree of symmetry along the *z*-axis, and there is no difference in the absorption of TE- or transverse magnetic (TM)-polarized light by normal incident light. In this work, TE-polarized light is selected as the incident light source.

The optical properties of the subwavelength-sized metamaterial absorber with sensitive responses to an electromagnetic field cannot be directly expressed by the optical parameters of tungsten and alumina. To better understand the high characteristics of the absorber in wide wavelength regions, the whole absorber is regarded as an effective medium. According to Equations (2), (3) and (5), the effective impedance at 280–4000 nm is calculated in detail based on impedance matching theory [39,40]. As illustrated in Figure 2b, the red and blue lines represent the variation in the real and imaginary parts of the effective impedance of the absorber with wavelength, respectively. When the impedance of the absorber matches with free space, it will achieve near-perfect absorption. The real part of the impedance of free space is Re(Z0)=μ(λ)/ε(λ)=1, and the imaginary part is Im(Z0)=0, which are represented by green and gray dotted lines, respectively. This means that when the real part of the effective impedance of the absorber is close to 1 (Re(Zeff)≈1), and the imaginary part is close to 0 (Im(Zeff)≈0), the absorber exhibits high absorption. It can be clearly observed from Figure 2b that Re(Zeff) and Im(Zeff), respectively, hover around the green and gray dotted lines in the wavelength range of 280–1700 nm, corresponding to extremely high absorptance of over 97.8% in the absorption spectrum. In the region beyond 2300 nm, the real and imaginary parts of the effective impedance vary drastically, which are far from the fixed dashed lines, exhibiting massive spectral reflections. Therefore, the selective absorption of the absorber is achieved remarkably.

As the spectral absorptance curve of the absorber shows in Figure 3, the absorber maintains more than 90% spectral absorptance in the 0.28–1.8 μm region, especially in the 0.28–1.7 μm region, which exhibits high spectral absorptance of over 97.8%, which almost perfectly covers most of the solar radiation range. We assume that the temperature of the absorber is 100 °C and the ambient temperature is 26 °C; according to the calculation of Equations (6)–(8), the total solar radiation absorptivity ($\alpha_{total}$) of the absorber under normal incident illumination is 97.1%, and only around 3% of the solar radiation energy is scattered. The total thermal emissivity ($\varepsilon_{total}$) above 2.5 μm is 8.5%, which is lower than the commercial application product criterion of 10%. A commercial solar-selective absorber should efficiently capture solar energy in the visible and near-infrared spectral regions while maintaining weak thermal emission in the mid- and far-infrared spectral regions. In other words, high absorptance ($\alpha_{total}$ > 0.9) over the spectral range of 0.28–2.5 μm is required. A non-selective absorber as a black body will lose the energy as thermal radiation in the IR region. Thus, it is also required to have low emittance ($\varepsilon_{total}$ < 0.1) in the spectral region of 2.5–30 μm [41].

The total photothermal conversion efficiency (η) reaches an astonishing 91.6%, and all data are listed in Table 1. Compared with other absorbers also based on metal tungsten [33,42,43], the MSSA proposed in this paper achieves a higher optical response in a wider wavelength band. In the case of similar structures, for instance, Ryu et al. present a material-versatile ultrabroadband absorber consisting of metal-coated self-aggregated Al_2_O_3_ nanowire bundles with multiscale funnel structures, which exhibits broadband absorption, with average absorption of 90% over a wavelength range of 300 to 2500 nm [44]. The absorber also achieves higher optical absorption in the same wavelength band. To avoid the formation of tungsten oxide (WO_3_) at around 300 °C, it is crucial to encapsulate the proposed solar absorber in a vacuum enclosure or deposit an ultrathin protective coating (SiO_2_) on the surface [45]. Although the thermal emissivity is slightly increased when the temperature of the absorber reaches 400 °C, the photothermal conversion efficiency still exceeds 90%, and thus the absorber has excellent practical prospects.

### 3.2. Underlying Mechanisms of the Broadband Absorption

To further reveal the physical mechanisms of the high optical response of the absorber in the ultraviolet, visible and near-infrared broadband range, six wavelength values at absorption peak positions were selected sequentially in the absorptance curve (302, 350, 449, 622, 968 and 1545 nm), and the electric field distribution in the x-z plane (|E|) and the magnetic field distribution in the y-z plane (|H|) were plotted after calculation.

As illustrated in Figure 4a, the strong electric field is mainly localized between adjacent tungsten metal truncated cones, especially at the bottom of the groove. The interaction between light and matter in metals is critically dominated by free electrons. When the incident light frequency is lower than the plasma frequency, the permittivity of the metal is negative. Moreover, the free electrons bound at the surface can collectively oscillate in the metal to form an extremely strong, localized electric field, which generates localized surface plasmon resonance (LSPR). The localized surface plasmon oscillations generated by adjacent truncated cones can couple with each other to further amplify the local electric field, forming hot spots in the electric field, as shown in Figure 4a [46]. On the one hand, the LSPR is primarily sensitive to the size of the micro–nano structure. In the design, the different spacings between the sidewalls of adjacent truncated cones are used to excite the LSPR corresponding to different wavelengths of light. In the electric field distribution diagrams at 302 and 350 nm, the electric field is mainly localized at the bottom of the metal groove. Meanwhile, in the map for 449 and 622 nm, the electric field distribution begins to move up along the sidewalls of the truncated cones, and the distance between adjacent truncated cones also increases. At the wavelengths of 968 and 1545 nm, the distance between adjacent truncated cones further broadens, and the local electric field distribution moves up further and occupies most of the sides. Significantly, when the interaction distance of the heterocharges excited along the sidewalls of the adjacent truncated cones becomes larger, the recovery coefficient of the response oscillating electrons will become smaller, which reduces the resonance frequency and thereby causes the redshift of the wavelength of the absorption peaks. Meanwhile, it can be noticed from the electric field distributions of 449 nm, 622 nm and 968 nm that the surface plasmon resonance coupling will also be excited on both sides of the same unit structure, due to the small distance between the upper parts of the two sidewalls in the same truncated cone. On the other hand, since the sidewall of the truncated cone is a curved surface, the metal surface charge distribution driven by the electric field is not uniform, which can excite multiple surface plasmon modes. Different modes can couple with each other, resulting in the enhancement of light absorption. Moreover, the inclined sides of the truncated cone are not perpendicular to the propagation direction of the normal incident light, and they produce multiple scattering effects on the incident light, which explains why the local electric field distributions become stronger as the height of the side decreases in the map at 968 nm and 1545 nm. In general, the cavity-localized surface plasmon resonance between adjacent truncated cones forms a light-trapping structure, which achieves a high spectral response in a wide range from the ultraviolet to the near-infrared region due to size effects and scattering. Additionally, as the wavelength of light increases, more and more light passes through the tungsten metal layer of 10 nm at the bottom of the truncated cones, and local electric fields are excited on the upper and lower metal surfaces to absorb light.

As depicted in Figure 4b, the magnetic field is mainly distributed on the top surface of the truncated cone and the middle MIM structure, and partially distributed on the sidewall surface of the truncated cone. When the electric field drives the electrons to gather in the *x*-axis direction, the local surface current will form in the *y*-axis direction, thereby generating magnetic field excitation on the surface, which is MPs [47]. As seen in Figure 4a, surface plasmon resonance coupling occurs in the upper part of the sidewall of the same truncated cone, thus generating the MPs on the circular plane at the top of the truncated cone, shown in Figure 4b. The magnetic field distribution on the top circular plan is strongest in the visible light region (400–760 nm), and the magnetic field excitation becomes gradually weaker in the near-infrared region. Moreover, the local surface current also appears on the sidewalls of the truncated cone to form weaker magnetic field excitation. It is well known that the electromagnetic field intensity decays exponentially while propagating into materials. The skin depth is δ=c/κω=λ/2πκ, where κ is the imaginary part of the refractive index (i.e., the extinction coefficient). The δ of tungsten is between 10 and 20 nm, which increases with wavelength [48]. The thickness of the upper layer of tungsten is set to 10 nm in the MIM structure; thus, the short-wavelength light will be reflected on the surface and absorbed by the sidewalls of the truncated cone by multiple scattering, while the long wave will enter the MIM optical resonator structure to generate magnetic field resonance and then be absorbed. Remarkably, there are two sources of long-wavelength photons: one is the photons of the incident light itself, and the other is a few lower-energy photons generated from the damping of the short-wavelength light by multiple scattering along the sidewall of the truncated cone. Consequently, some light enters the MIM structure and generates weak magnetic field resonance, as shown in the magnetic field distribution diagram at 302 nm and 350 nm. As the wavelength increases, more and more long-wavelength photons are absorbed in the MIM structure, which can be fully proven by the strong magnetic field distribution at 968 nm and 1545 nm in Figure 4b. Hence, the design of the MIM structure improves the absorption of the absorber in the near-infrared region.

In short, the underlying physical mechanisms of the wide-spectrum high absorption of the absorber can be attributed to the excitation of MPs on the surface of tungsten truncated cones, the cavity plasmon resonance that can form a light-trapping structure between truncated cones and the magnetic field resonance of MIM optical resonator structures. Additionally, another essential point is that the high-refractive-index imaginary component of metal tungsten shows the strong intrinsic absorption of light, which will be proven below. Therefore, the combination of multiple absorption mechanisms contributes to the excellent optical response of the absorber.

### 3.3. Geometric Effects on Spectral Absorption Performance at Normal Incidence

To further understand the interaction of this structure, obtain the optimized geometric parameters of the absorber and achieve the highest absorptance in solar radiation regions, the effects of geometric parameters were also investigated. We independently analyzed the individual geometry parameters while maintaining the rest of the parameters as constants.

#### 3.3.1. The Size Effect of Topmost W Truncated Cone

In our proposed plasmonic absorber, the truncated cones play a vital role in the enhancement of absorption, so we first studied the effect of the size of the truncated cone. As shown in Figure 5, the independent effects on the MSSA’s spectral absorptance of the diameter of the upper surface circle of the truncated cone *d*_1_, the diameter of the lower surface circle *d*_2_ and the height of the tungsten truncated cone *h* were calculated and analyzed, respectively. As seen in Figure 5a, as the diameter of the upper surface circle increases from 30 nm to 230 nm, the spectral absorptance decreases in the visible light region, the absorptance near 3 μm increases, and the cut-off wavelength remains basically unchanged. When *d*_1_ increases, the inclination of the sidewall of the truncated cone gradually increases, and then the distance to the upper part of the sidewall of the adjacent truncated cone decreases, so the excitation of the resonance mode of visible light in the upper part of the same truncated cone slightly attenuates, the degree of scattering on the sidewall of the cone decreases, and the coupled modes between the sidewalls of the adjacent truncated cones are also reduced, which together lead to weaker absorption. Moreover, the distance between the middle part of the adjacent truncated cones does not change too much, so the position of the cut-off wavelength does not change significantly. Due to the heterogeneous distribution of surface charges on the curved surface of the truncated cone, the coupled modes are also excited between the remote truncated cones in the second-nearest neighbor, demonstrating an absorptance enhancement of 3 μm. Additionally, when *d*_1_ is 30 nm, the top magnetic field excitation near 1.1 μm decreases because of the weakened size effect, and the absorptance drops slightly.

As seen in Figure 5b, as the diameter of the lower surface circle decreases from 400 nm to 150 nm, the overall spectral absorptance decreases, and the cut-off wavelength is obviously blueshifted. The reason is that as *d_2_* decreases, the inclination of the side walls of the truncated cone gradually increases, and then the distance between the side walls of the adjacent truncated cones increases, so the truncated cone tends to be a cylinder with a reduced size. As a result, the degree of light scatterings on the side walls of the cone decreases, and the coupling modes between the sidewalls of the adjacent truncated cones are also reduced. Consequently, the absorption decreases, and the cut-off wavelength is blueshifted. As depicted in Figure 5c, when the height of the tungsten truncated cone decreases, the amount of light scattering on the sidewall of the cone and the coupling modes between the adjacent sidewalls of the truncated cone will decrease, which is manifested as a broader decrease in spectral absorptance and the blueshift of the cut-off wavelength.

#### 3.3.2. The Effect of the Thickness of the Top W Layer and the Intermediate Al_2_O_3_ Layer in the MIM Structure

The sandwich-like MIM resonator structure obtains the optical response in the infrared region through the resonance of the optical field between the upper and lower metal surfaces. First, to enable the near-infrared light to enter the resonant cavity through the upper metal layer, the thickness of the layer was adjusted, as shown in Figure 6a. The absorbers with thinner (2 nm and 6 nm W) and thicker upper layers (14 nm and 18 nm W) both exhibit lower spectral absorptance, yet the absorber with the intermediate-thickness layer (10 nm W) presents the best spectral absorptance performance, which is because, when $t_{1}\approx \delta /2$, the light with a short wavelength is reflected in the upper metal layer but the light with a long wavelength enters the optical cavity. To obtain an ultrathin tungsten layer with uniform thickness, atomic layer deposition (ALD) can be considered as a suitable technique [45]. Secondly, the resonance wavelength of the MIM structure can be controlled by adjusting the thickness of the intermediate Al_2_O3 dielectric layer. As seen in Figure 6b, when *t*_2_ is 50 nm, 70 nm and 150 nm, the absorption of the structure in the infrared region decreases, and when *t*_2_ is 90 nm, 110 nm and 130 nm, the absorption of the structure in the infrared region is very close. In addition, when amplifying the absorption spectrum of the structure at 0.28–1.2 μm, which can be seen from the small figure in the upper-right corner, when $t_{2}=90\;\mathrm{nm}$, the entire absorption performance is better. In summary, the proposed MIM structure improves the absorption of infrared light above 1 μm.

#### 3.3.3. The Effect of the Period

Through the above analysis, it can be determined that the truncated cone is the core of the absorption. The distance between the adjacent truncated cones affects the absorption performance of the MSSA in the visible and infrared regions. The variation in the period substantially changes the distance between adjacent truncated cones. As shown in Figure 7a, as the period of the unit cell expands from 400 nm to 800 nm, the visible light absorption of the absorber decreases at around 0.5 μm, while the absorption in the infrared region of 1.2–2.3 μm is also reduced. It is known, from the discussion of Figure 4a, that the increase in the distance between adjacent truncated cones reduces the surface plasmon resonance coupling effect, which can explain why the optical response weakens. When the period is large enough to reach 700 and 800 nm, the absorption characteristics of the MIM occupy the dominant position, showing resonance absorption around 1 μm, and the absorptance curve is consistent with the golden curve of the MIM structure in Figure 7b. The absorptance spectra of the tungsten thin film, MIM structure, pure tungsten truncated cone absorber and MSSA were investigated and are depicted in Figure 7b. Due to the high-refractive-index imaginary part of tungsten, the pure W film shows high absorption in the visible and near-infrared bands. In contrast, the absorption of the MIM structure in the near-infrared part is further improved, and the pure tungsten truncated cone absorber has an enhancement in absorption in the visible and near-infrared regions. Subsequently, the proposed absorber combines the multiple physical mechanisms of the natural absorption of tungsten metal, the high absorption of the MIM structure in the near-infrared region and the wide spectral response of the truncated cone structure, and demonstrates broadband high absorptance of over 95% in the 0.28–1.75 μm region.

### 3.4. The Effect of Polarization Angle and Oblique Incident Angle

To obtain as much solar radiation energy as possible, the proposed absorber should be polarization-insensitive in the event that the sunlight scattered by the atmosphere to the ground has an arbitrary polarization state. As shown in Figure 8, the absorptance spectra of the absorber at normal incidence angles of 0.28–4 μm under different polarization angles of 0–90° were studied in detail. Clearly, the absorber can consistently absorb light of any polarization state. Figure 9 shows a cross-sectional view of the electric field distribution at λ=500 nm and z=500 nm. When incident light with different polarization angles is incident on the absorber, the cross-section of the electrical field distribution rotates with the corresponding angle and presents a similar distribution of mode and intensity. Therefore, the polarization insensitivity of the absorber is derived from the unique design of the structure. The truncated cone surface microstructure proposed in this paper is a simple tetragonal primitive symmetry structure, which can eliminate the dependence on the polarization state of incident light.

In practical applications, sunlight has different incident angles at different times during the day, so the wide-angle absorption characteristics of different polarization states were also analyzed in the simulation. In Figure 10, the absorptance spectra of TE, TM and unpolarized light at oblique incidence angles of 0°–76° are displayed, respectively, wherein the absorption of unpolarized light was calculated as the average value of TE and TM absorption, which is A(Unpolarized)=A(TE)/2+A(TM)/2 [49]. The absorber maintains high absorption at an incident angle up to 65°. In particular, the average spectral absorption of TE and TM incident light in the 0.28–2.5 μm region reached approximately 88% and 93% at 65°, so the average spectral absorption of unpolarized incident light remains above 90%. Meanwhile, for an incident angle above 65°, the scattering of light by the top surface is enhanced, the effects of plasmon resonance in the microstructure become weak, and the average spectral absorption is maintained at only around 70–80%. Thus, the simulations further serve to demonstrate the enhanced broadband optical absorption absorber with excellent omnidirectional optical properties.

## 4. Conclusions

In summary, a high-efficiency, broadband and plasmonic nanostructure solar absorber consisting of tungsten truncated cones and a tungsten–alumina–tungsten MIM structure has been systematically proposed and numerically investigated. The proposed structure has a spectral region from 280 nm to 1700 nm, with spectral absorptance greater than 97.8%. The total solar radiation absorptivity of the designed solar absorber in the ultraviolet, visible and near-infrared regions reaches 97.1% at normal incidence, while the total normal thermal emissivity at wavelengths larger than 2.5 μm is lower than 0.085 at 373.15 K, and high photothermal conversion efficiency of 91.6% is also achieved. Additionally, the physical mechanisms responsible for the high broadband absorptance are attributed to the synergistic effect of the MPs on the surface of tungsten metal, the cavity plasmon resonance between truncated cones, the magnetic field resonance of the MIM optical resonator and the inherent loss of metal tungsten, which result in the impedance matching with the free space. Furthermore, FDTD simulations were also performed to investigate the geometric effects of the size of the topmost W truncated cone, the thickness of layers in the MIM structure and the period. Moreover, the absorber enables the optical abilities of the independent of the polarization angle and strong absorption for a wide range of incident angles up to 65°. Overall, we believe that the absorber presented here can provide a direction in the design of broadband solar absorbers based on multiple plasmonic resonance.

## Figures and Tables

**Figure 1 nanomaterials-12-04456-f001:**
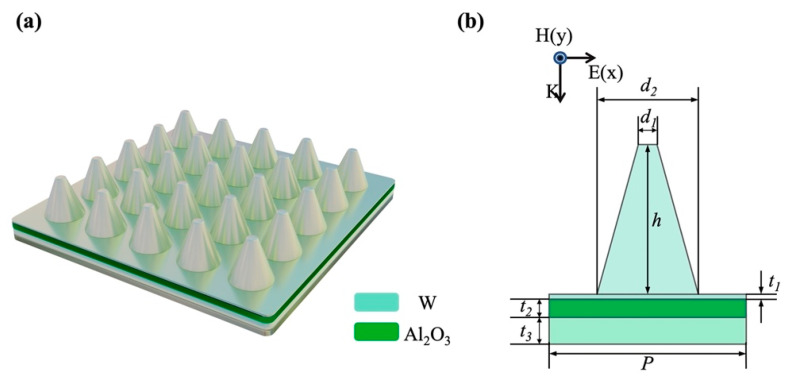
Schematic diagram of the MSSA: (**a**) three-dimensional schematic of the MSSA, (**b**) two-dimensional cross-section schematic of the unit cell structure.

**Figure 2 nanomaterials-12-04456-f002:**
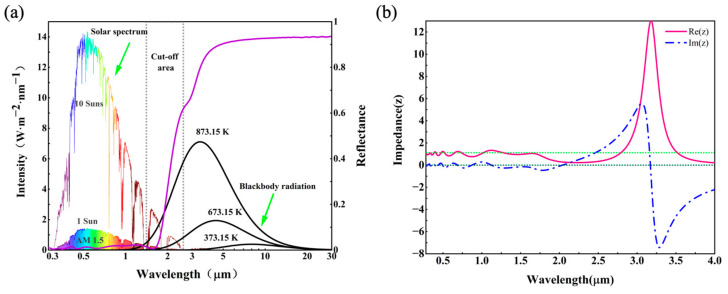
(**a**) Solar spectrum (AM 1.5) at 1, 10 optical concentrations, black body radiation spectrum (black line) at 373.15 K, 673.15 K, 873.15 K interspersed with reflection spectrum (purple line) of the sample and cut-off area (light grey region). (**b**) The calculated real and imaginary parts of the effective impedance for the proposed MSSA.

**Figure 3 nanomaterials-12-04456-f003:**
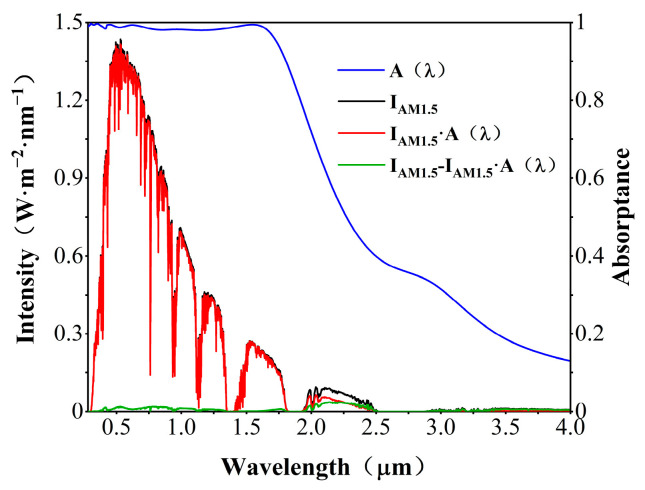
Solar spectral irradiance AM 1.5 of global tilt, absorption spectrum of proposed meta-structure solar absorber and residual solar irradiance in the ultraviolet, visible and near-infrared regions (0.28–4 μm).

**Figure 4 nanomaterials-12-04456-f004:**
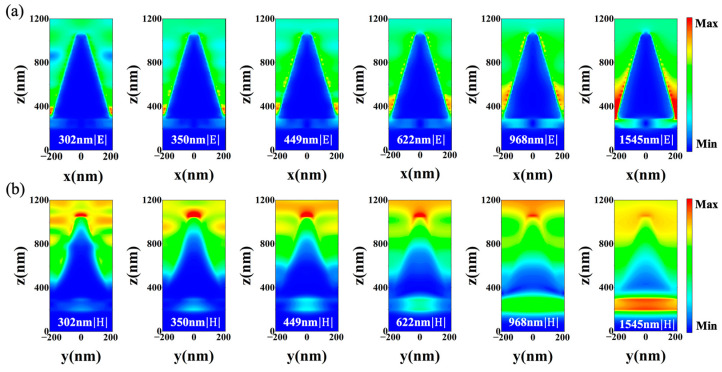
Calculated distributions of the normalized electric field (|E|) and magnetic field (|H|) in the x-z plane and y-z plane at wavelengths of 302 nm, 350 nm, 449 nm, 622 nm, 968 nm and 1545 nm, respectively. (**a**) Electric field in x-z plane. (**b**) Magnetic field in y-z plane.

**Figure 5 nanomaterials-12-04456-f005:**
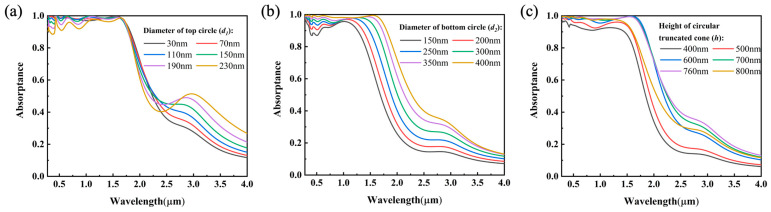
Energy dissipation within W truncated cone. (**a**) The contributions of the diameter (*d*_1_) of the top circle. (**b**) The contributions of the diameter (*d*_2_) of the bottom circle. (**c**) The contributions of the height (*h*) of W truncated cone.

**Figure 6 nanomaterials-12-04456-f006:**
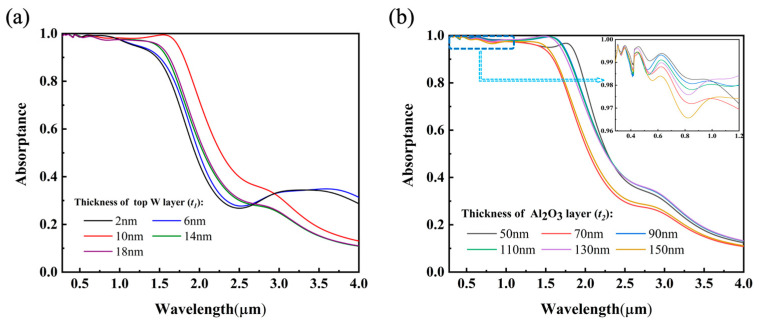
(**a**) Absorptance spectra of absorbers when the thickness (*t*_1_) of the top W layer varies from 2 to 18 nm. (**b**) Absorptance spectra of absorbers when the thickness (*t*_2_) of the intermediate Al_2_O_3_ layer varies from 50 to 150 nm.

**Figure 7 nanomaterials-12-04456-f007:**
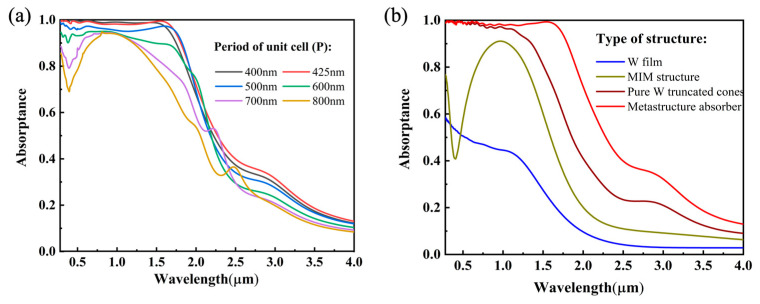
(**a**) The effect of periods (*P*) on the absorptance of the meta-structure absorber. (**b**) Absorptance spectra comparison of the simple W film, MIM structure, pure W truncated cone absorber and meta-structure absorber.

**Figure 8 nanomaterials-12-04456-f008:**
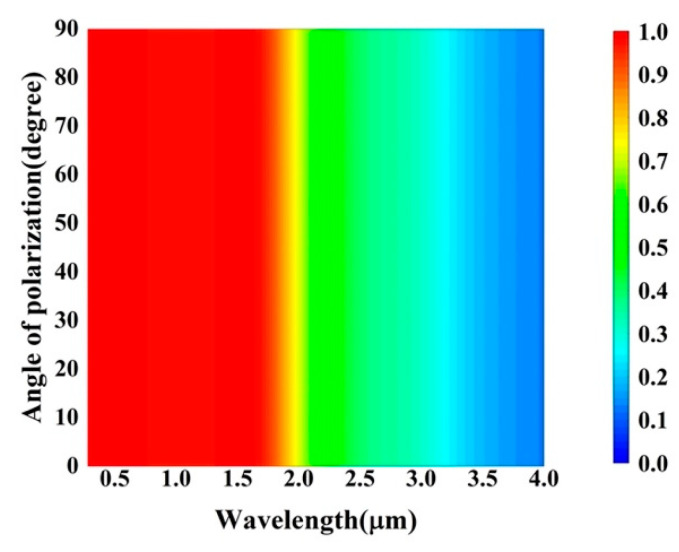
Absorptance spectra of the present absorbers with different angles of polarization at normal incidence.

**Figure 9 nanomaterials-12-04456-f009:**

Simulated electric field distribution of the meta-structure absorber at 500 nm with polarization angles of 0°, 30°, 60° and 90°.

**Figure 10 nanomaterials-12-04456-f010:**
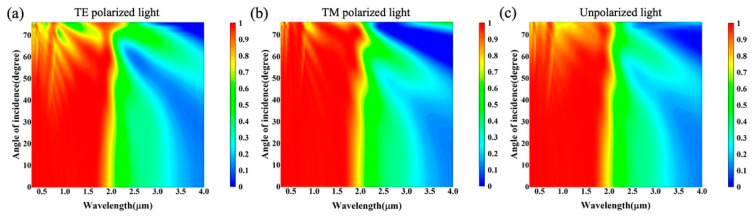
Absorptance spectra of absorbers with different incident angles. (**a**) TM-polarized wave. (**b**) TE-polarized wave. (**c**) Unpolarized light. The incident angle step is 2° from 0° to 76°.

**Table 1 nanomaterials-12-04456-t001:** The total solar radiation absorptivity, total thermal emissivity and solar thermal conversion efficiency for the proposed absorber at different operating conditions.

Operating Temperature (K)	Solar Concentration Ratio C	$\alpha_{total}$	$\varepsilon_{total}$	η
273.15	1	0.971	0.085	0.916
673.15	20	0.971	0.122	0.903

## Data Availability

Not applicable.

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
