# Peer review of "Plasmonic Nanostructures for Broadband Solar Absorption Based on Synergistic Effect of Multiple Absorption Mechanisms"

_nanomaterials, 2022, doi:10.3390/nano12244456_

Round 1

Reviewer 1 Report

The work of Sun et al. concerns the design of nanostructure plasmonic wide angle absorbers. The authors succeeded so some extent to demonstrate the high efficiency obtained of the designed structure based on W-oxide-W. However, their work was not supported or compared to any potential experimental work.

1) Hence I suggest they provide with experimental comparison of the same or analogue structure showing similar performances.

2) Please revise this statement page 2 line 53 : "which resulting in the enhancement of thermal emission". Did you mean "reducing the energy conversion efficiency by thermal emission

3) Page 2 line 80, authors can build in this experimental citation to provide with experimental results (question 1). Please precise here briefly the fabrication route and the type pf coating equipment

4) Authors need to define the abbreviation at first citation then keep using only the abbreviation, this has to be revise in the whole mansucript

5) Page 7 line 229, precise de figures number and connect the figure description with the right figure number such as Figure 4a...

Author Response

Dear Reviewer,

Thank you for your carefulness and conscientiousness. Your suggestions are really valuable and helpful for revising and improving our paper. According to your suggestions, we have made the following revisions on this manuscript:

  1. Hence I suggest they provide with experimental comparison of the same or analogue structure showing similar performances.

Response: Thank you very much for your valuable comments, we have added the experimental comparison of the same or analogue structure showing similar performances on page 6 lines 290-295.

  1. Please revise this statement page 2 line 53: "which resulting in the enhancement of thermal emission". Did you mean "reducing the energy conversion efficiency by thermal emission"?

Response: This understanding is correct. The enhancement of thermal emission will lose more energy, resulting in a lower absorption efficiency. To avoid misunderstanding, we have made some adjustments shown on page 2 line 74 of the revised version.

  1. Page 2 line 80, authors can build in this experimental citation to provide with experimental results (question 1). Please precise here briefly the fabrication route and the type pf coating equipment.

Response: Thank you very much for your constructive comments, we have added the details of the fabrication route. Briefly, tungsten and alumina layers can be respectively deposited by magnetron sputtering and reactive magnetron sputtering, then the surface structures can be fabricated by electron beam etching or using a tightly focused femtosecond laser beam seen on page 2 lines 104-110.

  1. Authors need to define the abbreviation at first citation then keep using only the abbreviation, this has to be revise in the whole manuscript.

Response: Thank you very much for your valuable comments, we have made the adjustment in the new revised version.

  1. Page 7 line 229, precise de figures number and connect the figure description with the right figure number such as Figure 4a.

Response: Thank you very much for your valuable comments. This whole paragraph analyzes the absorption mechanism through the electric field distribution in Figure 4 (a), so the pictures refer specifically to 6 diagrams in Figure 4 (a). We have made the adjustment on page 7 line 359.

Thank you again for your valuable comments and suggestions. We look forward to hearing from you at your earliest convenience.

Sincerely yours,

Dingquan Liu  PhD, Prof.     

Director of Dept. of Optical Thin Film, Material and Device 

Shanghai Institute of Technical Physics, Chinese Academy of Sciences
500 Yutian Road, Hongkou Distract, Shanghai, 200083 China
E-mail: dqliu@mail.sitp.ac.cn
Tel: +86-21-25051303,  Mobil: +86 15216701692

Junli Su  

Shanghai Institute of Technical Physics, Chinese Academy of Sciences
500 Yutian Road, Hongkou Distract, Shanghai, 200083 China
E-mail: sujl@shanghaitech.edu.cn
Mobil: +86 18621145860

Reviewer 2 Report

In  this work, Su et al. introduce a comprehensive theoretical study on the performance of plasmonic solar selective absorbers based on the combination of a nanostructured W film and a metal-oxide-metal resonator.

The field of research is of timely interest, especially because nanostructured selective absorbers for solar energy conversion have been thoroughly studied in the last decade, and are most likely ready to be considered for use on a large scale, provided that scalable and feasible fabrication technologies are available. In this sense, the work presented by the authors gives undoubtedly a significant contribution.

The manuscript is well-written, and the used methodology is rigorous. The work is well-organized, and results are clearly presented. However, in my opinion, there are some points to be revised before considering the paper suitable for publication, as well as some information to be added.

I’ll be more detailed in the following general comments:

1)     In the introductory section, authors forgot to mention one of the most important application of plasmonic solar selective absorbers, i.e. their use in thermionic solar energy converters, both pure (TE) and photon-enhanced (PETE). In this case, nanostructured surfaces greatly enhance solar absorptance and reduce thermal emittance, allowing for both an efficient absorption of solar photons in the UV-Visible range and an efficient heating of the cathode, so to enhance thermionic emission from the cathode surface. I suggest to mention this in the introductory section with adequate references (see Advanced Energy Materials 8, 1802310 (2018) and Applied Surface Science 380, 8-11 (2016)).

2)     Authors mention metals as the most common surface plasmon materials for the fabrication of solar selective absorbers. However, the best results in terms of absorptance in the solar spectrum have been obtain with the so-called “black” semiconductors, such as black Silicon (see Energy Environ. Sci.7, 3223-3263 (2014)) and black Diamond (see Carbon 138, 384-389 (2018)), showing solar absorptance values up to 99.1%. It should be mentioned in the introductory section, for completeness.

3)     As for the fabrication technologies of plasmonic solar selective absorbers, author should mention LIPSS (i.e., Laser-Induced Periodic Surface Structures), which have been thoroughly investigated in the past decade, and are induced by femtosecond laser treatment of the absorber surface (see Optical Materials 107, 109967 (2020) and Carbon 105, 401-407 (2016)). The use of femtosecond laser may not require necessarily vacuum as the electron beam lithography (EBL) technique mentioned by the authors, and does not need for time-consuming technological steps as EBL. I suggest to elaborate a little bit more on this in the introductory section, by mentioning LIPSS as a well-established type of solar selective absorbers.

4)     Lines 52-53. Authors state that “the electron-electron scattering during the noble metal plasmon resonance also produces excessive Joule heat, which resulting in the enhancement of thermal emission”. In case of absorbers coupled to thermionic emitters, however, this could be an advantage, because enhanced thermal emission may result in a higher heat-to-current conversion efficiency. I suggest to mention this particular case.

5)     The theoretical study is really well-described and presented by the authors. However, there is poor uniformity in the terms used to identify the same quantity. For instance, authors sometime use “absorbance”, some other “absorptance”, or “absorptivity”, or even “absorption”, or again “absorption rate” to indicate the same thing. The same goes for “emittance” and “emissivity”. Again, the same goes for “conversion rate” and “conversion efficiency”. All of these terms have a precise definition. The value given in the abstract (97.8%), for instance, is an “absorptance” value. Pay attention to their correct use in the paper, both in the text and the figures.

6)     Line 128. Write “Planck” and not “Plank”.

7)     Line 204. Authors write that “The total thermal emissivity above 2.5 μm is 8.5%, which is 10% lower  than the specified thermal emissivity of the application product.” To what application product do they refer? It should be better explained, or at least a reference should be provided.

8)     Figure 5. Textboxes should be enlarged. Readability is poor.

9)     Line 433. Replace “solar absorptance efficiency” with “solar absorptance”.

10) Even if it's a theoretical study,  I suggest to elaborate a little bit more on the feasibility of the proposed structure, in particular in the case of the thin metal film acting as the upper layer of the MIM structure. Authors write that the thickness of this layer is in the range 2-18 nm. Therefore, it is an extremely thin layer, and there are very few techniques available for the growth of such a thin W film. In most cases, indeed, the film rugosity would be higher than the thickness of the film itself, resulting in a non-uniform thickness, or even in a non-continuous film. Probably the only suitable technique is atomic layer deposition (ALD), and it should be mentioned.

11)  A few words should be spent on the maximum operating temperature of the proposed device before possible degradation. For instance, if authors propose their device for concentrating solar power applications, they surely know about the high temperature involved. Tungsten oxide WO3 starts to form at about 300°C. To avoid problems like this, it is therefore crucial to encapsulate the proposed solar absorber in a vacuum enclosure, in my opinion.

Author Response

Dear Reviewer,

Thank you for your carefulness and conscientiousness. Your suggestions are really valuable and helpful for revising and improving our paper. According to your suggestions, we have made the following revisions on this manuscript:

  1. In the introductory section, authors forgot to mention one of the most important application of plasmonic solar selective absorbers, i.e. their use in thermionic solar energy converters, both pure (TE) and photon-enhanced (PETE). In this case, nanostructured surfaces greatly enhance solar absorptance and reduce thermal emittance, allowing for both an efficient absorption of solar photons in the UV-Visible range and an efficient heating of the cathode, so to enhance thermionic emission from the cathode surface. I suggest to mention this in the introductory section with adequate references (see Advanced Energy Materials 8, 1802310 (2018) and Applied Surface Science 380, 8-11 (2016)).

Response: Thank you for your constructive comments, after research about the two articles, “Optimization of black diamond films for solar energy conversion” and “Solar Thermionic-Thermoelectric Generator (ST2G): Concept, Materials Engineering, and Prototype Demonstration”, we found such schemes also significantly improve the efficiency of solar energy utilization. This broadens our horizons of the application method of broad-spectrum solar selective absorbers. We have cited the relevant articles seen lines 31 and 32.

  1. Authors mention metals as the most common surface plasmon materials for the fabrication of solar selective absorbers. However, the best results in terms of absorptance in the solar spectrum have been obtain with the so-called “black” semiconductors, such as black Silicon (see Energy Environ. Sci.7, 3223-3263 (2014)) and black Diamond (see Carbon 138, 384-389 (2018)), showing solar absorptance values up to 99.1%. It should be mentioned in the introductory section, for completeness.

Response: Thank you very much for your valuable comments, after research about the two articles, “Black silicon: fabrication methods, properties and solar energy applications” and “Optimization of black diamond films for solar energy conversion”, we believed that so-called “black” semiconductors construct the surface topography to form the light-trapping structures, which increase the scattering of incident light in the structure, thus reduce the surface reflection, however, the intrinsic absorption of the semiconductor does not cover a wider band in nature, we prefer to call it light-trapping enhancement absorption of surface textured “black” film seen lines 36 and 37.

  1. As for the fabrication technologies of plasmonic solar selective absorbers, author should mention LIPSS (i.e., Laser-Induced Periodic Surface Structures), which have been thoroughly investigated in the past decade, and are induced by femtosecond laser treatment of the absorber surface (see Optical Materials 107, 109967 (2020) and Carbon 105, 401-407 (2016)). The use of femtosecond laser may not require necessarily vacuum as the electron beam lithography (EBL) technique mentioned by the authors, and does not need for time-consuming technological steps as EBL. I suggest to elaborate a little bit more on this in the introductory section, by mentioning LIPSS as a well-established type of solar selective absorbers.

Response: Thank you for your insightful comment, indeed, the laser-induced periodic surface structures (LIPSS) treatment may not require necessarily vacuum and does not need for time-consuming technological steps as EBL. We've learned about LIPSS long before this work from the two articles, “Enhanced selective solar absorption of surface nanotextured semi-insulating 6H–SiC” and A 90-nm-thick graphene metamaterial for strong and extremely broadband absorption of unpolarized light”. We feel uncertain about whether LIPSS can handle refractory metals like tungsten.Anyway, LIPSS could be considered a feasible approach to form the nanostructure seen lines 104-110. And we are looking forward to further discussion with you about LIPSS technique.

  1. Lines 52-53. Authors state that “the electron-electron scattering during the noble metal plasmon resonance also produces excessive Joule heat, which resulting in the enhancement of thermal emission”. In case of absorbers coupled to thermionic emitters, however, this could be an advantage, because enhanced thermal emission may result in a higher heat-to-current conversion efficiency. I suggest to mention this particular case.

Response: Thank you for your constructive comments. We have mentioned the particular case seen lines 74-76.

  1. The theoretical study is really well-described and presented by the authors. However, there is poor uniformity in the terms used to identify the same quantity. For instance, authors sometime use “absorbance”, some other “absorptance”, or “absorptivity”, or even “absorption”, or again “absorption rate” to indicate the same thing. The same goes for “emittance” and “emissivity”. Again, the same goes for “conversion rate” and “conversion efficiency”. All of these terms have a precise definition. The value given in the abstract (97.8%), for instance, is an “absorptance” value. Pay attention to their correct use in the paper, both in the text and the figures.

Response: Thank you very much for your valuable comments. This was our big mistake, we have made adjustments both in the text and the figures in the revised version。

  1. Line 128. Write “Planck” and not “Plank”.

Response: Thank you very much for your valuable comments and we have corrected this embarrassing spelling mistake.

  1. Line 204. Authors write that “The total thermal emissivity above 2.5 μm is 8.5%, which is 10% lower than the specified thermal emissivity of the application product.” To what application product do they refer? It should be better explained, or at least a reference should be provided.

Response: Thank you very much for your valuable comments, we have added a section to explain the commercial application product criterion seen lines 282-288.

  1. Figure 5. Textboxes should be enlarged. Readability is poor.

Response: Thank you very much for your valuable comments. We have enlarged textboxes in Figure 3,5,6,7.

  1. Line 433. Replace “solar absorptance efficiency” with “solar absorptance”.

Response: Thank you very much for your valuable comments and we have change the words.

  1. Even if it's a theoretical study,  I suggest to elaborate a little bit more on the feasibility of the proposed structure, in particular in the case of the thin metal film acting as the upper layer of the MIM structure. Authors write that the thickness of this layer is in the range 2-18 nm. Therefore, it is an extremely thin layer, and there are very few techniques available for the growth of such a thin W film. In most cases, indeed, the film rugosity would be higher than the thickness of the film itself, resulting in a non-uniform thickness, or even in a non-continuous film. Probably the only suitable technique is atomic layer deposition (ALD), and it should be mentioned.

Response: Thank you very much for your valuable comments. We have added the content about atomic layer deposition (ALD) technique, seen lines 503 and 504.

  1. A few words should be spent on the maximum operating temperature of the proposed device before possible degradation. For instance, if authors propose their device for concentrating solar power applications, they surely know about the high temperature involved. Tungsten oxide WO3starts to form at about 300°C. To avoid problems like this, it is therefore crucial to encapsulate the proposed solar absorber in a vacuum enclosure, in my opinion.

Response: Thank you very much for your valuable comments. To maintain the solar thermal conversion efficiency above 90%, the operating temperature of the absorber is about 400°C. Therefore, it is necessary to prevent the formation of tungsten oxide WO3. Your suggestion on encapsulating the proposed solar absorber in a vacuum enclosure is very practical. There is another possible way to avoid problems like this,which is depositing a ultrathin protecting coating (SiO2) by atomic layer deposition (ALD) on the surface, seen lines 298.

Thank you again for your valuable comments and suggestions. We look forward to hearing from you at your earliest convenience.

Sincerely yours,

Dingquan Liu  PhD, Prof.     

Director of Dept. of Optical Thin Film, Material and Device 

Shanghai Institute of Technical Physics, Chinese Academy of Sciences
500 Yutian Road, Hongkou Distract, Shanghai, 200083 China
E-mail: dqliu@mail.sitp.ac.cn
Tel: +86-21-25051303,  Mobil: +86 15216701692

Junli Su  

Shanghai Institute of Technical Physics, Chinese Academy of Sciences
500 Yutian Road, Hongkou Distract, Shanghai, 200083 China
E-mail: sujl@shanghaitech.edu.cn
Mobil: +86 18621145860

Reviewer 3 Report

The present manuscript is very interesting, the analysis is very well written and the results are supported by the data. Therefore I accept for publication the paper in its present form.

Author Response

Dear Reviewer,

Thank you for your recognition of our work. According to the rest reviewers’ suggestions, we have made some minor revisions on this manuscript.We look forward to hearing the following guidance from you at your earliest convenience.

Sincerely yours,

Dingquan Liu  PhD, Prof.     

Director of Dept. of Optical Thin Film, Material and Device 

Shanghai Institute of Technical Physics, Chinese Academy of Sciences
500 Yutian Road, Hongkou Distract, Shanghai, 200083 China
E-mail: dqliu@mail.sitp.ac.cn
Tel: +86-21-25051303,  Mobil: +86 15216701692

Junli Su  

Shanghai Institute of Technical Physics, Chinese Academy of Sciences
500 Yutian Road, Hongkou Distract, Shanghai, 200083 China
E-mail: sujl@shanghaitech.edu.cn
Mobil: +86 18621145860
